# Understanding the Families’ Perceptions of Adapted Physical Activity for Individuals with Autism Spectrum Disorder through Metaphors

**DOI:** 10.3390/healthcare11020267

**Published:** 2023-01-15

**Authors:** Bekir Erhan Orhan, Aydın Karaçam, Ali Selman Özdemir, Eda Gökçelik, Alpar Aser Sabuncu, Laurențiu-Gabriel Talaghir

**Affiliations:** 1Faculty of Sport Science, Istanbul Aydın University, Beşyol Mah.Inönü Street, no 38, 34295 Istanbul, Türkiye; 2Faculty of Physical Education and Sport, Dunărea de Jos University, Garii Street, no 63-65, 800003 Galati, Romania; 3Institute of Sport, Tourism and Service, South Ural State University, Sony Krivoy Street, no 60, 454080 Chelyabinsk, Russia

**Keywords:** adapted physical education, physical activity, special needs, special education

## Abstract

The aim of this study was to understand the perceptions, understanding and experiences of the families of individuals with autism spectrum disorder (ASD) related to adapted physical activity (APA), and their educators, through their use of metaphors. The research was based on systematic content analysis in the qualitative research model. The data were collected based on metaphors. The participants included 85 families of individuals with ASD attending private institutions operating in Istanbul and Ankara. The metaphors used by the families were examined under two headings: adapted physical activities, and educators. When the metaphors used to describe APA were examined, four themes emerged, education, emotion, support and development, and these themes were divided into the categories of guidance, skill, affection, and care. When the metaphorical perceptions of educators were examined, three themes emerged: education, emotion, and social adaptation. These were further categorized as experience, knowledge, entertainment, independence, happiness, and treatment. The metaphors showed that experiences in APA support increased self-confidence for individuals with autism spectrum disorder and support the social adaptation of individuals who have the opportunity to apply knowledge and experience. This research shows that families have positive perceptions of APA educators and their lessons.

## 1. Introduction

Autism spectrum disorder (ASD) is a neurodevelopmental disorder of neurobiological origin that affects both the development of communication and social interaction skills due to the presence of repetitive and restricted behaviours, lack of interest in general activities, speech development issues, difficulties with motor movement and the use of objects [1]. Social communication deficits may include impairments in attention span and social reciprocity, as well as challenges in using verbal and nonverbal communication for social interaction. They also have little flexibility when there are changes to their routine, and hypersensitivity and/or hyposensitivity to sensory information [2]. Although control over motor skills is not part of the diagnostic criteria for ASD, studies have shown that individuals with ASD frequently appear to have issues with gross motor [3] and object control and often deficiencies in fine motor skills [4,5,6].

All people, whatever their situation, have the right to a quality education that allows them to develop the skills necessary to function adequately and independently in the world around them [7,8]. Individuals with ASD face several problems when accessing education, some of them have to do with the characteristics of their condition and others with the conditions of the school and the context. When we talk about the problems related to the conditions of the school and the context, we refer to the difficulties of access to the educational institution, the physical conditions of the school, access to the common curriculum, the training of teachers to attend to individuals with special educational needs, and to the general training of educational personnel on the diagnosis. Physical education and physical activities are educational tools through which individuals with ASD can improve their current motor performance levels and gain new skills [9,10].

Borremans [11] states that “promoting physical exercise and activity through physical education is very important for everyone, but mainly for individuals with special needs such as ASD”. The benefits of regular physical activity support the importance of physical exercise in improving overall health and well-being. In research on physical exercise, it is known that physical activity reduces stress, improves general health, improves motor skills, and helps to improve individual morality, self-confidence, discipline, responsibility, sociability, communication, friendship, and coordinated work. Considering the existing physical, mental, and social difficulties faced by individuals with ASD, it is seen that they need special education programs to improve their health and quality of life [12]. At this point, adapted physical education and sports activities emerge as educational tools to meet the needs of individuals with ASD.

Adapted physical activity (APA) is a program designed to meet the specific needs of individuals. In relation to the changing special needs and skills of individuals, they may show less progress in specific fields and in developmental characteristics compared to their peers with normal development and offer specially selected activities and programs for them [13]. The purpose of APA is not physical education practices for a certain group but to meet the needs of individuals with differences by applying physical education. In other words, it is not to provide a different physical education for the elderly, pregnant women and the disabled, but to provide adaptive physical education services according to individual differences. APA is a broad term that aims to collate all areas of intervention not covered by formal physical activity. In addition, according to DePauw and Doll Tepper [14], APA is defined as “all physical activities and sports in which special attention is paid to the interests and abilities of the disabled, individuals with health problems, or the elderly, with restrictive conditions”. APA can be considered an interdisciplinary body of knowledge dedicated to identifying and resolving individual differences in physical activity and adapting to improve them. APA includes, but is not limited to, physical education, sports, entertainment, dance and the creative arts, nutrition, medicine, and rehabilitation [15]. Planning special events that support motor development and include lifelong learning for students with special needs within the scope of general education enables us to focus on APA.

Researchers often ask teachers, parents, and individuals who help in the teaching process about their experiences with individuals with ASD [16]. This is a way of examining how individuals with ASD perceive an experience. Considering the difficulties individuals with ASD have in expressing themselves and their feelings, it is a more appropriate method to evaluate the process through the opinions of individuals responsible for their education and care.

In this context, metaphor studies offer people who are responsible for caring for these individuals the opportunity to express themselves. A metaphor represents more than its original meaning. It creates new meanings that did not exist before [17]. Thanks to metaphors, mental connections are established between unrelated things, while the underlying meanings of the concepts remain unchanged [18]. According to Lakoff and Johnson [19], conceptual metaphors structure our thinking. They enable the transfer of meaning from one object to another based on perceived similarity. According to Şaban [20], the concept of “like” in metaphor studies is generally expected to provide information about the connection between the subject of the metaphor and its source. The concept of “because”, which is asked later, aims to explain the reason why the metaphor is expressed in this way. According to Morgan [21], “the use of metaphor means a way of thinking and a way of seeing that permeates our understanding of the world in general”. In this respect, a metaphor is a powerful mental construct that an individual can employ in understanding and explaining a highly abstract, complex, or theoretical situation.

When metaphor studies on individuals with ASD were examined, no metaphorical study was found to have learned the views of families regarding adapted physical activity, and for this purpose, it was aimed that families with ASD individuals could clarify the participant’s perceptions, understanding and experiences of the inner world of the APA applications used in the education of these individuals through metaphors.

## 2. Materials and Methods

The research was based on semantic content analysis in the qualitative research model. “Content analysis” is the technique that allows the content of “communications” to be investigated by classifying the manifest elements or contents of said communication or message into “categories”. According to Bernard Berelson [22], “Content analysis is a research technique for the objective, systematic and quantitative description of the manifest content of communication”. Another well-known definition is, according to Ole Holsti [23], “any research technique that serves to draw inferences by systematically and objectively describing certain features in a text”. In the definition made by Klaus Krippendorff [24], “Content analysis is a research technique [a set of users] for making reproducible and valid inferences from a text (data to data binding)” is in the same methodological line.

### 2.1. Data Collection

In the collection of research data, families of individuals with ASD were asked to create reasoned metaphors about APA and the educators who gave this adapted physical activity. In this direction, among the participants, “Adapted physical activity is like …; Because …”, “Educators who give adapted physical activity are like … Because …”. It is located in the last part of the form, there is also information about demographic characteristics. Before the data were collected the families were informed about the metaphors, their understanding was tested with sample metaphors, and the data collected face to face.

Before starting the research, ethics committee approval was obtained from Istanbul Aydın University, Social and Human Sciences Scientific Research and Publication Ethics Committee. During the data collection phase, the families of individuals with ASD were informed about the study, and then they filled out the forms themselves. The study was conducted in accordance with the Declaration of Helsinki, and the protocol was approved by the Ethics Committee of Istanbul Aydin University no. 2022/9 on 26 May 2022.

### 2.2. Data Analysis

Analysis of qualitative data: descriptive analysis was used in the analysis of metaphors created by families with ASD regarding APA and their trainers. Descriptive analysis is the presentation of data to the reader by adhering to the original form of the collected data as much as possible and by quoting directly what the individuals participating in the research said when necessary [25]. Descriptive analysis is mostly used in research in which the conceptual structure of the research is clearly determined beforehand [26]. The purpose of descriptive analysis is to present the findings to the reader in an organized and interpreted form. The data obtained for this purpose are first described in a systematic and clear way. After the descriptions made later are explained and interpreted, quotations are often included in order to clearly reflect the views of the individuals interviewed or observed. Adhering to this basic understanding, the conceptual structure of the research was determined by adopting the descriptive analysis approach, and the metaphor was supported by quoting directly from the participants’ thoughts regarding the relevant metaphor [27].

### 2.3. Validity and Reliability Study

Reporting the collected data in detail and explaining how the results were obtained are among the important criteria of validity in qualitative research [28]. In order to ensure the validity and reliability of this research, two basic procedures were carried out. First of all, the data analysis process was explained in detail in order to ensure validity, and all the data obtained were given together both quantitatively and qualitatively in the findings. Then, the themes and categories were presented to 3 experts and compared with the metaphor table created to ensure reliability. By determining the number of consensus and disagreement in the comparisons, the reliability of the research was calculated using the formula (Confidence = consensus/consensus + disagreement) of Miles and Huberman [29]. The result of the reliability calculation of the research is 94%. According to Miles and Huberman [29], the study is considered reliable if there is 90% or more consensus among researchers and experts in qualitative studies. In this case, it can be said that this study is reliable.

In the content analysis, the data obtained through the documents were analysed in four stages: (1) coding the data, (2) finding the codes, categories, and themes, (3) organizing the codes, categories, and themes, and (4) defining and interpreting the findings [29,30].

### 2.4. Coding of Data

The collected data were analysed, the data set was divided into meaningful sections, and it was established to which concept each section corresponds. Each section, which creates a meaningful structure, is given a name. During the data coding process, the data set was read several times and the coding process was carried out by repeatedly returning to the data set to process the coding process in a dynamic way [31,32,33].

### 2.5. Determination of Categories and Themes

Based on the codes discovered in the first stage of qualitative data analysis, it is necessary to identify the themes that can explain the data set at a more general level and collect the codes under certain categories. In this context, more abstract coding was involved in thematic coding, and firstly, the codes discovered in the first stage were brought together and the common features between them were determined [26,27,28,29,30,31,32,33,34]. Thematic coding is the determination of the similarities and differences of codes with different characteristics. Therefore, related codes are grouped together. This grouping is a categorization process. Categories of the same type make up themes. Internal consistency is an important condition that should be considered while doing thematic coding. For this reason, the degree of establishing a meaningful relationship with the data set on which the determined themes are based was considered in thematic coding. In addition, since the degree to which all the determined themes can explain the data obtained in the research in a meaningful way, external consistency, is important, the determined themes are separate from each other, but after the categories and themes are determined they can form a meaningful whole within themselves, the coding process and the data are organized according to the codes [33,35,36].

### 2.6. Organizing Data by Codes, Categories and Themes

As a result of detailed coding and thematic coding, a systematic structure was created to organize the collected data in a meaningful way. Then, based on this structure, the collected data were reorganized, and it was necessary to reorganize the data, and in some cases to provide detailed coding and thematic coding. During the data editing phase, pioneering findings were reached and the data were redefined and interpreted according to these findings. It is important to describe, explain, and present the data in a language that the reader can understand. For this reason, for the information in the findings section that was focused on, at this stage, the researcher did not include his own views and comments and presented the collected information to the reader in a processed form [27,29,34,37].

The last stage, as seen in Figure 1, gave meaning to the collected data and explained the relationships between the findings, establishing cause–effect relationships, drawing conclusions from the findings, and establishing the importance of the results obtained [38,39].

### 2.7. Participants of the Study

The research participants consisted of 85 families of individuals with ASD who are attending 2 private education institutions operating in the Üsküdar district of Istanbul, the Yenimahalle district of Ankara, and those providing APA for individuals with ASD. The study group were selected from Ankara, the capital of Turkey, and Istanbul, the most populous city, where adapted physical education lessons are actively provided. While quoting the views of the participants, a coding system (“P1, P2, P3”, etc.) which expressed that they were participants was preferred.

#### 2.7.1. Parent’s Data

As seen in Table 1, 74.1% of the parents participating in the study were female and 25.9% were male, 83.5% of them are married and 16.5% of them are divorced. A total of 64.4% of parents do not receive the support of a third person, relative or employee, while 30.6% receive the support of a third person.

Table 2 shows that only 1 of the parents has never received any education, 7 of them have master’s/doctorate degrees, and the number of parents who have primary school, high school, and university education is similar.

#### 2.7.2. Data of Individuals with ASD

From Table 3, it can be seen that the main age distribution is in the 7–10 age range and in the 11–14 age range. From this age range distribution, it can be said that the majority of individuals with ASD who attend classes are individuals who have just started school.

When Table 4 is examined, it is seen that 58.8% of the individuals attending the courses are male and 41.2% are female.

Table 5 shows that the majority of individuals with ASD, 71.8%, do not engage in any physical activity outside of APA.

A total of 37.6% of the individuals with ASD in the participating families have been attending classes for more than 5 years according to Table 6.

## 3. Results

When the perceptions of the participating families about educators are examined, 4 themes emerged as education, emotion, support and development. These 4 themes are again divided into the categories of guidance, skill, care and affection (see Figure 2).

The majority of metaphors were related to education and emotion as shown in Table 7. It indictes the metaphorical perceptions of the participating families about “educators” and the 4 common themes which emerged were education, emotion, support and development. These 4 themes are again divided into guidance, skill, affection, and care.

When the theme of “Education” is examined, it consists of 28 codes from 2 categories. When these 4 categories are examined, it is seen that families have a positive opinion about the educators who manage the APA training, the teaching methods of the educators, and how they guide their children. In this direction, the codes are divided into guiding and skill categories.

Examples of the metaphors established by families in the theme of education are given below.

P1; “He/She who guides is like a hero, because he/she teaches everything.”

P3; “She/he is like a sculptor because they shape my child.”

P18; “She/he is like a friend, because they teach my child through love.”

P30; “She/he is like our friend, because they easily teach our child the things that we have difficulty with.”

P50; “She/he is like a computer mouse because they choose the right lessons.”

When the “Emotion” theme is examined, it consists of 23 codes from 3 categories. When these 3 categories are examined, it can be said that although the care and affection categories are intense, skill is also an important factor. When the emotion category is examined, it can be said that the affection, interest, and closeness of the educators to their children through the metaphors used by the families in this theme come to the fore, but it can be said that the spinners use their experience skills while forming an emotional bond with the children.

Some metaphors established by families in the theme of emotion are given below.

P7; “She/he is like a comrade, because she is with my child throughout the entire education process.”

P13; “She/he is like a member of the family because they show affection like a member of the family.”

P48; “She/he is like a perfume, because their positivity spreads its beauty to everyone around them.”

P62; “She/he is like an artist because it makes my child love dance and music.”

P74; “It’s like having fun in the park because kids love them and have fun.”

When the “Support” theme is examined, it consists of 20 codes from 2 categories. When the codes collected under the guidance and care categories are examined, it can be said that the families see the educators as a guide, support and sincere.

Some metaphors established by families in the theme of support are given below.

P36; “She/he is like a favourite aunt or uncle because she/he is with you from when you take your first step.”

P37; “She/he is like a light, because they illuminate the way for my child to walk.”

P53; “She is like a mother because of the constant support they give through education.”

P57; “She/he is like a family member; they give as much support and effort as we do.”

P64; “She/he is like an angel because she/he does everything they can for my daughter.”

When the “Development” theme is examined, it consists of 14 codes from 2 categories. When these 2 categories are examined, it is seen that the metaphors are concentrated in the skill category. Considering that one of the basic building blocks of development is the appropriate use of skills and abilities, it is quite natural for families to make metaphors for the skill category. In addition, a guide category was created.

Some metaphors established by families in the theme of development are given below.

P28; “They are like friends because they are in harmony with them in any situation.”

P32; “It is like a door key, because it is the key to the door that will open for my child to live a better life.”

P39; “She/he is like a psychologist because they calm my child down.”

P70; “She/he is like a fairy godmother/godfather, because she/he adds goodness and beauty to the life of every child they work with.”

P73; “She/he is like a hero because she/he is an idol who gives skills.”

P78; “She/he’s like a witch because my son is someone else beside her/him.”

P79; “It’s like a spice because it does what we can’t with variety.”

P85; “She/he is like a tree because they develop roots with the soil and grow together.” 

When the perceptions of the participating families about APA are examined, 3 themes emerged as education, emotion, and social adaptation. These 3 themes are again divided into the categories of experience, knowledge, entertainment, independence, happiness, and treatment (see Figure 3).

Examining the distribution of categories in Table 8, it is seen that the majority are connected to social adaptation and emotion. When the “Education” theme is examined, it consists of 15 codes from 2 categories. When these 2 categories are examined, it is seen that the knowledge and experience categories appear predominantly. It is seen that the families think that the individuals who attend the classes both have fun and gain experience through APA.

Some metaphors utilised by families in the theme of education are given below.

P4; “It’s like a favourite lesson because it teaches everything about life.”

P20; “It is like a pomegranate because it has a lot of hidden gains.”

P35; “It’s like life because they learn everything.”

P49; “It is like a field because knowledge grows.”

P51; “It’s like school because he/she goes there to learn.”

P56; “It is like the branches on a tree because there is a lot of information.”

When the “Emotion” theme is examined, it consists of 31 codes from 2 categories. When these 2 categories are examined, it is seen that the happiness and entertainment categories appear predominantly. The importance for families is having a good time in APA classes.

Common metaphors used by families on the theme of emotion are given below.

P34; “It’s like an amusement park because it’s a lot of fun.”

P42; “It’s like a game because it loves to play.”

P44; “It is like home, because it is comfortable and happy.”

P58; “It’s like the excitement of the first day of primary school because he/she’s very curious about school.”

P63; “It is like an amusement park because they are mentally very happy.”

P69; “It’s like rollerblading because they need balance and control to be comfortable.”

When the “Social Adaptation” theme is examined, it divides into 39 codes from 3 categories. When these 3 categories are examined, it is seen that the experience and independence categories appear predominantly. Regarding the adapted physical activity, it is seen that the families think that the individuals who attend the classes are supported to become independent through the experience. It is seen that the category of treatment has emerged in line with the metaphors used by families in these categories.

Some metaphors used by families in the theme of social adaptation are given below.

P32; “It is like a door because it opens new places for her/him to live a better life.”

P37; “It is like the road, because it is the place where she/he walks to live comfortably.”

P38; “It’s like a test, because when you get it right, it makes you happy.”

P46; “It is like medicine, because the obedient became a person who started to solve his/her own problems.”

P47; “It’s like a door key, because every skill he learns makes him/her a little more free, individual, independent.”

P48; “It’s like a compass because it helps you find direction.”

P52; “It is like a life-jacket because he/she has overcome his/her fear of water.”

P61; “It is like the theatre, because they take on a completely different character while watching and practicing in the lessons.”

P68; “It’s like a puzzle, because all pieces need to fit together to work as one.”

P73; “It’s like therapy because it is a safe, non-judgemental and free space.”

P77; “It’s like a clock, because it makes it more capable.”

P81; “He/she is like a saviour because he/she helps him when he/she is struggling.”

P84; “It is like water because it is transparent.”

## 4. Discussion

The aim of this study is to represent the opinions of the families of individuals with ASD about APA and the educators who administer it through their use of metaphors. Today, we are all aware of the importance of health in society and that physical activity produces a positive mood with many other health benefits, but it is of particular importance to evaluate it from the perspective of families who can closely observe positive or negative changes. The benefits of regular and moderate exercise have been known for a long time and supported by many scientific studies [40,41]. These benefits are not just seen physically but are also apparent in the psychology of participating individuals, and so APA is fundamental for the rehabilitation of people with severe physical disabilities [42]. It is known that these activities not only help to improve the physical capacities of any person but also have mental and sensory benefits, in addition to being effective in people who experience loss of performance due to physical and mental conditions such as illness, accident, and trauma [43].

Although all individuals have the right to an education, often just going to school can be a major challenge for those with ASD. Individuals with ASD often have sensory dysfunction, so things like bright lights, peers shouting, or the sound of the school bell may be the dominant stimuli that trigger excessive anxiety, aggression, or behaviours such as self-harm [44]. These individuals need an individualized program that must be compatible with the right and need of all students to receive a quality education shared with their peer group. When we consider the definition of educator in general, the level of knowledge of educators and the ability to accurately convey this knowledge to individuals draws attention as a very important criterion. Considering that the perception and expression of the emotions and desires of these individuals will improve their integration at both the personal and social level, innovations and individualization in education should be made to detect and treat possible abnormalities and deficiencies in understanding the emotions of these individuals [45]. To the greatest extent possible, educators should understand and appreciate the feelings and needs of each person with special needs and their family.

Educators have core goals such as personalizing support, adapting to each person, considering personal options, determining goals and priorities together based on the needs of the individual, supporting personal development, and acquiring functional and important skills. The educators act decisively and in a positive sense [46]. If we evaluate in general, we can say that the families feel the educators are close to them and their children. Considering the problems experienced by individuals with ASD in expressing their feelings and empathizing, it can be said that the formation of an emotional bond with both the individuals attending the classes and their families positively affects the participation of individuals with ASD in APA. According to McLeskey and Waldron [47], the aim of successful education should be to support in a way that is a “natural and modest” part of education in the classroom. For these individuals with social communication problems, increasing their self-confidence and feeling happier, socializing, and minimizing any social communication problems which may have emerged as a result of ASD.

In these individuals, education should be offered as early as possible and should focus on the acquisition of daily life skills as well as social, communication, academic, and behavioural development. It is very important for these individuals, who have problems in socialization, to transform the new knowledge they have acquired into experiences, to show them what they can achieve. It is important to provide education in the least restrictive educational environments to these individuals who are faced with many restrictive behaviours in society and feel emotionally deficient because of these restrictive situations. The fact that these individuals who have a limited number of interaction areas feel more comfortable through these lessons stands out as an important output for these individuals who have difficulty in expressing themselves.

Helping everyone develop their full potential and life goals can be achieved with appropriate support. Through these activities, different factors such as independent action, self-confidence, acceptance by others, socialization, and communication can be supported by contributing to the health and quality of life of individuals [48]. Socialization is a constant concern for parents and an expressed desire of children [46]. It can be said that these individuals started to act independently in society in line with the opinions of their families, thanks to the experiences they gained in the lessons. It emerges because of the metaphors of families, where knowledge, experience and social adaptation of individuals who can apply these experiences in structured lessons are supported by increasing their self-confidence.

## 5. Conclusions

When the metaphorical perceptions of the families of individuals with ASD towards APA and educators are examined, four themes emerged: education, emotion, support, and development. It can be said that the theme of education seems dominant, but as well as educating the individuals the teachers provide emotional support and meet sensory needs, contributing to their development in a positive way. Although these are the characteristics that every educator should have, it should not be forgotten that individuals with ASD need more than individuals with typical development. [49]. The opinions of the families are that the educators guide the individuals with special needs in a way that equips them with skills they can use in class and out, they transfer skills and knowledge. In addition to the importance of the educator’s skills, the affection and interest in the individuals with ASD come to the fore. Individuals with ASD lack perception of emotions, which is an important element in emotional development. It can be said that parents clearly emphasize the importance of the interest shown by educators as well as their guidance, they see the educators as close to them and care about their suggestions and thoughts. It is seen that the knowledge and abilities of the educator come to the fore. It was observed that the parents emphasized that the educator should guide the individuals with ASD correctly, and they appreciated the skills of the educator. To make a general evaluation, apart from the educator having a certain skill set, it is important to use the necessary teaching skills correctly and appropriately in the studies of individuals with ASD. In line with the opinions of the families, it can be said that APA provides the opportunity for these individuals to practice skills needed for everyday life.

As a result of the metaphors that families have used for educators, it can be concluded that they generally think the skills of educators who are close to the ASD individuals are enough, especially when there is an emotional bond between both the family and the educator–student. When individuals with ASD feel a bond, they develop more meaningful real-life tools through APA. For ASD individuals experiencing emotional and behavioral problems, the bond formed with educators becomes more important [50]. The metaphors used by parents about the educators show that the families see the APA educators as close to them, often as a family member, during the education process, they are a major influence on the lives of their children. In this respect, it can be concluded that educators care about the duration of education and their students in line with the opinions of the families. It is seen as a positive variable in terms of education, especially when working with social communication problems.

When the metaphors used by the families participating in the research for APA were examined, three themes emerged: education, emotion, and social adaptation. In line with the metaphors used by families, it was concluded that individuals with ASD, just like any other student, gained new knowledge and experiences through lessons when having fun, and this helps them to act independently and with increased self-confidence. This APA is seen as important for individuals with ASD who do not as easily express their emotions and may experience difficulties in communication [49,50,51]. These individuals, when having fun, learn more and feel happier in themselves. It can be said that through these lessons, individuals with ASD minimize the problems they experience due to their current disadvantages and the negative emotions and experiences resulting from their inability to experience life fully.

Parents’ satisfaction with programs for their children varies greatly depending on the student’s special needs [52]. Many studies tend to focus only on the needs of the child and family views are ignored. Considering family views can be a step towards more effective relationships and similar programs for students with ASD or special needs. If families think that the programs are supportive and responsive to their needs, they will tend to take part in more programs and be more involved themselves in their child’s education [53].

The findings obtained from the study are important because they highlight the important contribution adapted physical activity makes to the lives of individuals with ASD. It is seen that the ASD individuals’ families believe that APA helps individuals with ASD to fulfill their daily life skills adequately and supports their integration into society. This can be seen from the metaphors they have used to describe APA and educators. In future studies, the participation of individuals with ASD in APA classes can be supported, new participation opportunities can be offered and developed.

## Figures and Tables

**Figure 1 healthcare-11-00267-f001:**
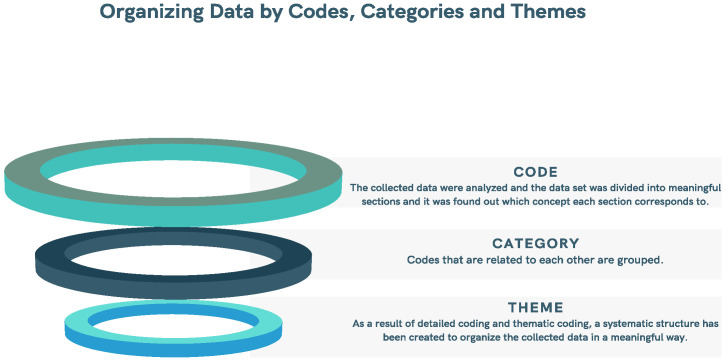
Organizing data by codes, categories, and themes.

**Figure 2 healthcare-11-00267-f002:**
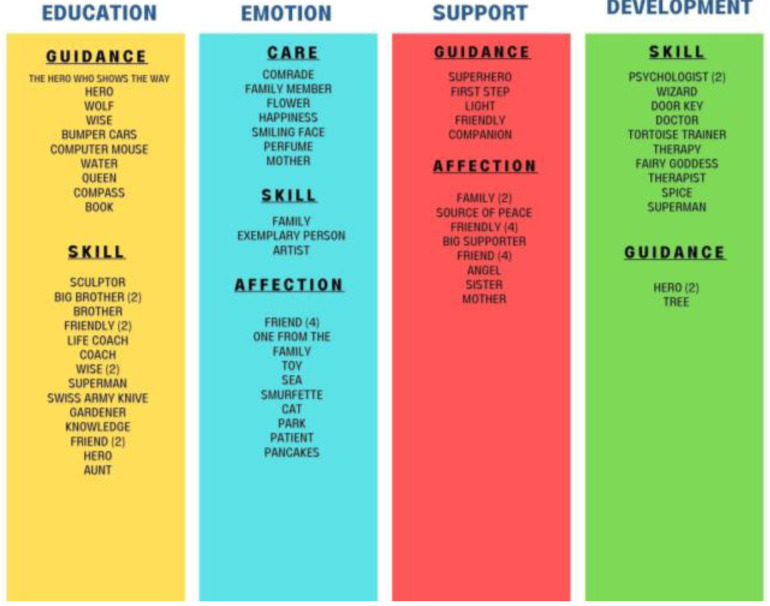
Parents’ metaphors about educators. Themes, categories, and codes that emerged because of the metaphors produced by the families participating in the research for “educators” are presented in Figure 2.

**Figure 3 healthcare-11-00267-f003:**
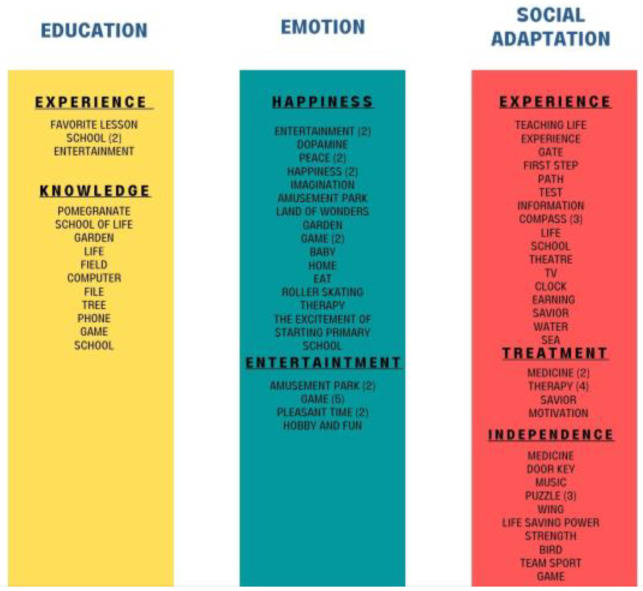
Parents’ metaphors about APA. Themes, categories, and codes that emerged because of the metaphors produced by the families participating in the research for “educators” are presented in Figure 3.

**Table 1 healthcare-11-00267-t001:** Marital status of the parents, gender distribution of parents, and parents receiving support from a third person in the care of individuals with ASD.

		Frequency	Percent
Gender	Woman	63	74.1
	Male	22	25.9
	Total	85	100
Marital Status	Married	71	83.5
	Divorced	14	16.5
	Total	85	100
Responsible 3rd Person	No	59	69.4
	There is	26	30.6
	Total	85	100

**Table 2 healthcare-11-00267-t002:** Parents’ educational status.

Education Status	Frequency	Percent
Primary	22	25.9
High	33	38.8
University	22	25.9
Master’s/Ph.D.	7	8.2
None of them	1	1.2
Total	85	100.0

**Table 3 healthcare-11-00267-t003:** Age distribution of individuals with ASD who received APA.

Age	Frequency	Percent
2–6 Years	12	14.1
7–10 Years	27	31.8
11–14 Years	20	22.8
15–18 Years	9	10.7
19+ Years	17	20.3
Total	85	100.0

**Table 4 healthcare-11-00267-t004:** Gender distribution of individuals with ASD who received APA.

Gender	Frequency	Percent
Woman	35	41.2
Male	50	58.8
Total	85	100.0

**Table 5 healthcare-11-00267-t005:** Physical activities that individuals with ASD participate in other than APA they receive.

Physical Activity	Frequency	Percent
No	61	71.8
Swimming	13	15.3
Skate	2	2.4
Ballet	2	2.4
Basketball	2	2.4
Gymnastics	2	2.4
Table tennis	1	1.2
Run	1	1.2
Bike	1	1.2
Total	85	100.0

**Table 6 healthcare-11-00267-t006:** Participation time of individuals with ASD in APA.

Duration	Frequency	Percent
1 Year	11	12.9
2 Years	3	3.5
3 Years	12	14.1
4 Years	17	20.0
5 Years	10	11.8
5+ Years	32	37.6
Total	85	100.0

**Table 7 healthcare-11-00267-t007:** Distribution of categories and codes for the metaphors that families produced about educators’ APA.

Theme	Categories	Codes (N)	Codes Total (N)
Education	Guidance	10	28
	Skill	18	
Emotion	Care	7	23
	Skill	3	
	Affection	13	
Support	Guidance	5	20
	Affection	15	
Development	Skill	11	14
	Guidance	3	
		85	85

**Table 8 healthcare-11-00267-t008:** Distribution of categories and codes for the metaphors that families produced about educators about APA.

Theme	Categories	Codes (N)	Codes Total (N)
Education	Experience	4	15
	Knowledge	11	
Emotion	Happiness	21	31
	Entertainment	10	
Social Adaptation	Experience	19	39
	Treatment	8	
	Independence	12	
		85	85

## Data Availability

The data that support the results of this study are available from the corresponding author upon reasonable request.

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
