# Peer review of "Understanding the Families’ Perceptions of Adapted Physical Activity for Individuals with Autism Spectrum Disorder through Metaphors"

_healthcare, 2023, doi:10.3390/healthcare11020267_

Round 1

Reviewer 1 Report

The manuscript entitled "UNDERSTANDING THE FAMILIES’ PERCEPTIONS OF ADAPTED PHYSICAL ACTIVITY FOR INDIVIDUALS WITH AUTISM SPECTRUM DISORDER THROUGH METAPHORS" by Orhan et al. is one of the interesting articles on Family perceptions of children with ASD. 

The manuscript is well presented, and I have only minor suggestions about figures. 

Adding figure legends would be easy to understand the figures. 

Still, it is unclear that ASD children are associated with other intellectual disability diseases. 

Author Response

Comment: Adding figure legends would be easy to understand the figures. 

Answer: In addition to figures, tables that give numbers have been added.

Reviewer 2 Report

It might be published after minor revision.

In fact, the text is ready to be published in its present form.

Before forwarding their text to the editor, Authors may decide:

¾     Make minimal improvements (abstract, conclusions); the text will be ready for publishing but may cause some critics.

¾     Invest some effort and improve the text; the text will be ready for publishing.

¾     Invest some more effort in reconstructing indicated issues. The text will be a high-quality publication.

Please refer to the appended file, where:

A list of strengths is provided.

The list of defects is formulated.

Suggestions for improvements are given.

Author Response

¾     Make minimal improvements (abstract, conclusions); the text will be ready for publishing but may cause some critics.

¾     Invest some effort and improve the text; the text will be ready for publishing.

¾     Invest some more effort in reconstructing indicated issues. The text will be a high-quality publication.

Answer: All done whenever possible.

Reviewer 3 Report

Dear authors,

First of all, I would like to thank the authors of the manuscript for the effort they have put into the preparation of the manuscript entitled: Understanding the Families’ Perceptions of Adapted Physical Activity for Individuals with Autism Spectrum Disorder Through Metaphors.

I would be grateful if the authors would consider the following comments derived from my review of the manuscript:

ABSTRACT

1) The length of the abstract is much longer than allowed. The manuscript template on the Healthcare web page states: "A single paragraph of about 200 words maximum". In addition, the document also encourages authors to use the abstract structure they have used (background, methods, results and conclusions), but without headings. Authors are advised to make the appropriate modifications to adapt to the requirements of the journal.

2) It would be appropriate for abbreviations not to appear in the summary.

INTRODUCTION

3) The following appears on line 59: [4-5-6]. When more than three references are cited with consecutive numbering, the first and the last are joined by a hyphen. Consequently, it should read: [4-6].

4) I miss that the authors include some references in certain statements made in the introduction (for example, in what is provided in lines 63-71). Revise everything written in the introduction in this regard.

5) In relation to what was provided in comment number 4, more current references should also be included, since of the 18 mentioned in the introduction, only 3 have been published since 2018.

MATERIALS AND METHODS

6) I consider it advisable that the authors of the manuscript modify the structure of the "materials and methods" section to make it clearer. Although the authors may choose the structure, they consider most appropriate, the following is proposed:

2.1. Design and Subjects (incorporate the information from the paragraphs on lines 128-137 and 222-225)

2.2 Procedure

2.2.1 Instruments

2.2.2 Data Collection

2.2.3 Data Analysis (this section may integrate the sub-sections on: validity and reliability, data coding, determination of categories and themes, and organization of data by codes, categories and themes).

2.2.4 Ethical Approval (in addition to information from lines 146-149, information from lines 489-492 can be included).

7) How the information was collected should be explained in more detail: response time, instrument used, etc.

8) I would like to know if any software was used for data analysis. If it was used, it should be mentioned. 

RESULTS

9) Decimal numbers are written in English with a period, not a comma. For example, in table 1, the percentage of married fathers is 83.5

10) Tables 1, 2, 3 and 4 should be grouped into a single table.

11) Tables 5, 6, 7 and 8 should be grouped into a single table.

DISCUSSION

12) This section should be expanded by linking the results obtained and the existing scientific evidence. Inferences should also be drawn from the results obtained, for example, by trying to answer why the themes (education, emotion, support and development) may have been obtained.

13) The limitations of the study and future lines of research should be included at the end of the section.

CONCLUSIONS

14) I would like to congratulate the authors of the manuscript for the quality of this section.

I would like the authors of the manuscript to take into consideration all the comments made after the revision of the manuscript.

Author Response

The length of the abstract is much longer than allowed. The manuscript template on the Healthcare web page states: "A single paragraph of about 200 words maximum". In addition, the document also encourages authors to use the abstract structure they have used (background, methods, results and conclusions), but without headings. Authors are advised to make the appropriate modifications to adapt to the requirements of the journal.

 It would be appropriate for abbreviations not to appear in the summary.

Answer: The summary has been shortened and arranged as desired.

The following appears on line 59: [4-5-6]. When more than three references are cited with consecutive numbering, the first and the last are joined by a hyphen. Consequently, it should read: [4-6].

Answer: arranged as desired.

I miss that the authors include some references in certain statements made in the introduction (for example, in what is provided in lines 63-71). Revise everything written in the introduction in this regard.

Answer: Supported by source.

Ethical Approval (in addition to information from lines 146-149, information from lines 489-492 can be included).

Answer: Added.

How the information was collected should be explained in more detail: response time, instrument used, etc.

Answer: Added

I would like to know if any software was used for data analysis. If it was used, it should be mentioned. 

Answer: No software was used for data analysis.

Decimal numbers are written in English with a period, not a comma. For example, in table 1, the percentage of married fathers is 83.5

Answer: Fixed.

Tables 1, 2, 3 and 4 should be grouped into a single table.

Answer: Table 1,2,3 has been merged but not added because table 4 has too many variables.

Tables 5, 6, 7 and 8 should be grouped into a single table.

Answer: It wasn't merged because there were too many variables.

Round 2

Reviewer 3 Report

Dear Authors,

First of all, I would like to thank you for taking into consideration the comments I made in the first revision of the manuscript.

Second, I would like to add two comments to the second version of the manuscript:

1) Abbreviations in the introduction are not desirable. This comment was already made in the first revision. 

2) In the abstract there are some verbs that are in the present tense and should be in the past tense. For example, in the first line it says the following: "the aim of this study is....". The verb used should be in the past tense, i.e.: "the aim of this study was...". Review the entire abstract in this regard and make the appropriate changes.

I hope you will take these two comments into consideration to improve the quality of the manuscript entitled: Understanding the Families' Perceptions of Adapted Physical Activity for Individuals with Autism Spectrum Disorder Through Metaphors

Kind regards.

Author Response

First of all, thank you for your nice comments and suggestions.

Thanks to you, the article is in a better state. We corrected the part you mentioned in the abstract.

We also looked at the comments about the other abbreviations you made, but we couldn't do anything because we couldn't find out what the problem was. We left it the same as similar articles have corrections in the same style.